# The heritability of multi-modal connectivity in human brain activity

Giles L Colclough[1,2,3]*, Stephen M Smith[2], Thomas E Nichols[4,5], Anderson M Winkler[2], Stamatios N Sotiropoulos[2,6], Matthew F Glasser[7], David C Van Essen[7], Mark W Woolrich[1,2,3]*

[1]Oxford Centre for Human Brain Activity (OHBA), Wellcome Centre for Integrative Neuroimaging, Department of Psychiatry, University of Oxford, Oxford, United Kingdom; [2]Oxford Centre for Functional MRI of the Brain (FMRIB), Wellcome Centre for Integrative Neuroimaging, Nuffield Department of Clinical Neurosciences, University of Oxford, Oxford, United Kingdom; [3]Department of Engineering Science, University of Oxford, Oxford, United Kingdom; [4]Department of Statistics, University of Warwick, Coventry, United Kingdom; [5]Warwick Manufacturing Group, International Manufacturing Centre, University of Warwick, Coventry, United Kingdom; [6]Sir Peter Mansfield Imaging Centre, School of Medicine, University of Nottingham, Nottingham, United Kingdom; [7]School of Medicine, Washington University, St. Louis, United States

*For correspondence:
giles.colclough@ohba.ox.ac.uk (GLC);
mark.woolrich@ohba.ox.ac.uk (MWW)

**Abstract** Patterns of intrinsic human brain activity exhibit a profile of functional connectivity that is associated with behaviour and cognitive performance, and deteriorates with disease. This paper investigates the relative importance of genetic factors and the common environment between twins in determining this functional connectivity profile. Using functional magnetic resonance imaging (fMRI) on 820 subjects from the Human Connectome Project, and magnetoencephalographic (MEG) recordings from a subset, the heritability of connectivity among 39 cortical regions was estimated. On average over all connections, genes account for about 15% of the observed variance in fMRI connectivity (and about 10% in alpha-band and 20% in beta-band oscillatory power synchronisation), which substantially exceeds the contribution from the environment shared between twins. Therefore, insofar as twins share a common upbringing, it appears that genes, rather than the developmental environment, have the dominant role in determining the coupling of neuronal activity.
DOI: https://doi.org/10.7554/eLife.20178.001

## Introduction

Intrinsic human brain activity enables inference about the pathways and processes of information transfer in the brain. Studying intrinsic activity, when the brain is in a resting state, has given insights into many aspects of healthy and diseased brain function. Resting-state function is characterised by spatially separated regions organised into networks of strongly correlated activity (*Beckmann et al., 2005*; *Smith et al., 2012*). These networks represent both local connectivity and longer-range communication. Importantly, the strength of resting-state functional connectivity reflects many aspects of cognitive function. Connectivity in the brain changes throughout the life cycle: networks associated with attention and control may continue to develop in late adolescence (*Barnes et al., 2016*) and network integrity degrades during the ageing process (*Dennis and Thompson, 2014*). Many neurological diseases, including schizophrenia and Alzheimer's disease (*Sheline and Raichle, 2013*; *Greicius, 2008*), have been associated with major alterations to the strength and organisation of

functional connectivity (*Stam and van Straaten, 2012*; *Stam, 2014*; *van Straaten and Stam, 2013*). In healthy subjects, intrinsic brain activity can predict not only performance in a task, but also the specific regions which will show increased activity during that task (*Sala-Llonch et al., 2012*; *Yamashita et al., 2015*; *Zou et al., 2013*; *Smith et al., 2013*; *Tavor et al., 2016*). Furthermore, recent evidence suggests that the organisation of human brain function at rest is associated with a broad range of behavioural and life-style traits, and reflects a generalised measure of intelligence (*Smith et al., 2015*).

Here, we set out to identify the relative importance of genetic and shared environmental factors in determining these fundamental patterns of cortical communication. We employ functional magnetic resonance imaging (fMRI) recordings to perform a heritability analysis on the strength of functional connectivity within a network of 39 regions. Additionally, we perform the same heritability analysis on source-localised magnetoencephalographic (MEG) recordings (*Van Veen et al., 1997*) from a subset of the subjects. This complementary analysis allows us to focus specifically on communication mediated by the coupling (correlation) of oscillatory amplitudes, within particular frequency bands, using MEG as a more direct measure of neuronal activity that is unaffected by vascular confounds.

We present separate analyses of functional networks estimated from the fMRI response (*Smith et al., 2013*), and of MEG-derived networks in the theta (4–8 Hz), alpha (8–13 Hz) and beta (13–30 Hz) oscillatory bands. These bands span the frequency range within which the most convincing patterns of resting-state MEG connectivity have been shown to be expressed (*Hipp et al., 2012*; *Baker et al., 2014*; *Hillebrand et al., 2012*; *Brookes et al., 2011*; *Mantini et al., 2007*; *Marzetti et al., 2013*; *de Pasquale et al., 2012*; *de Pasquale et al., 2016*). We employ resting-state recordings from 820 subjects from the Human Connectome Project (HCP; *WU-Minn HCP Consortium et al., 2013*); these data are publicly available, and have undergone a standardised pre-processing procedure. The HCP is a twin study, and the subjects from the S900 data release with all fMRI (MEG) resting-state scans comprise 103 (19) monozygotic twin pairs, who share 100% of their genetic structure and a common environment, and 54 (13) dizygotic twin pairs, who share 50% of their genetic structure and a common environment. Each subject provided three MEG scans (of 6 minutes each), and four fMRI scans (of 15 minutes each). This stratified sample allows estimation of the relative effects of genetic influence and environmental factors on the variability observed in connectivity structure.

We compare the similarity in overall network structure between pairs of subjects, assessing the extent to which the functional connectivity of two subjects becomes more alike as their proportion of shared background and genetics increases. We fit variance-component models on each network edge, using recently developed permutation and bootstrap-based methods for fast, accurate, non-parametric statistical inference with family-wise control of type I errors (*Chen, 2014*). This provides estimates of the mean genetic and shared environmental influences on the observed phenotypic variation in cortical connectivity. We run similar heritability analyses on the oscillatory power and BOLD variance within the regions of interest (ROIs) which constitute the network nodes, to determine whether the observed effects of genes and developmental environment on cortical connectivity could be attributable to simple differences in signal strength. Additionally, we analyse the heritability of cortical folding patterns in each ROI, to investigate whether genetic control of anatomical structure contributes to the heritability of connectivity.

## Results

The functional connectivity measured with fMRI was estimated among 39 functionally defined cortical regions of interest (network nodes) by computing partial correlations between BOLD time courses representing each region. ROIs were spatially contiguous, covered both hemispheres, and were generated using fMRI from HCP subjects—see Materials and methods. The functional connectivity measured with MEG, corresponding to amplitude coupling of oscillations in each frequency band, was assessed between the same set of regions by correlating fluctuations in oscillatory power (*Engel et al., 2013*; *O'Neill et al., 2015*). Potential confounds to these MEG functional connectivity estimates induced by the leakage of source-reconstructed signals were reduced using a multivariate orthogonalisation technique (*Colclough et al., 2015*). These methods represent established practice for determining networks of functional connectivity in the two modalities (*Varoquaux and*

*Craddock, 2013*; *Smith et al., 2011*; *Smith et al., 2013*; *Brookes et al., 2012*; *Hipp et al., 2012*; *Colclough et al., 2016*; *de Pasquale et al., 2012*; *Hillebrand et al., 2012*). The group-averaged functional networks are shown in *Figure 1A* and rendered in 3D in the supplementary videos.

The fMRI connectome shows strong bilateral connectivity, together with fronto-parietal correlations and coupling between the anterior cingulate and posterior cingulate cortices, reflecting well-known patterns of resting-state networks in fMRI (*Beckmann et al., 2005*; *Smith et al., 2012*). In the MEG data, the alpha band shows a strongly connected visual system, with strong connectivity from the visual cortex extending out to the temporal and parietal lobes, and a highly coupled posterior cingulate. The beta-band exhibits strong bilateral coupling across the sensorimotor cortices, with connectivity continuing through the superior parietal lobes and down to the occipital cortex. The theta band also exhibits visual and motor connectivity. These patterns of connectivity are in general agreement with previously reported results in these frequency bands (*Hillebrand et al., 2012*; *Brookes et al., 2011*; *Baker et al., 2014*; *de Pasquale et al., 2012*; *de Pasquale et al., 2016*; *Hipp et al., 2012*; *Marzetti et al., 2013*; *Mantini et al., 2007*; *Colclough et al., 2015*).

*Figure 1B* (left-hand plot, middle row) shows that the structure of fMRI and MEG networks of functional activity are progressively more similar as the strength of relationship is increased, from unrelated subjects, through siblings and dizygotic twins to monozygotic twins ($p<10^{-3}$ in each case, except for $p^{\alpha}_{MZ<DZ} = 0.003$). (Although siblings and dizygotic twins share the same proportion of genetic material, siblings have different ages and inevitably share less similar environments than do twins. This may account for the significant increase in network similarity from siblings to dizygotic twins.) Also displayed, for comparison, is the distribution of network similarity over repeated recording sessions within the same subject. Tests for significance were performed by assessing the difference in mean of the logarithm of network separation, relative to the mean separation, using a non-parametric permutation-based $t$-test. All 21 tests performed for this paper are quoted after a false discovery rate correction for multiple comparisons.

A three-component variance model was fitted for the variability observed in the strength of coupling in individual edges. This model ascribes proportions of the variance in a phenotype, $\sigma^2$, either to additive shared genetics (A), shared environmental factors (C), or to measurement error and other known sources of variance (E),

$$\sigma^2 = A + C + E.$$

We estimate each of these components as a fraction of the total variance, giving $h^2 = A/\sigma^2$, known as the *heritability* of a trait, $c^2 = C/\sigma^2$ and $c^2 = E/\sigma^2$. These components are related by the identity

$$h^2 + c^2 + e^2 = 1.$$

Estimates of the mean heritability and contribution from common environment over these connections were constructed with bootstrapped confidence intervals, and the significance of the mean heritability was tested using non-parametric permutation-based methods. Although there is a large amount of variability in the data which is not attributable to either of these factors, we found that additive genetics accounted for 17%, on average, of the functional connectivity measured with fMRI. In MEG, 19% of the variability in oscillatory coupling strengths in the beta band, and 8% in the alpha band, was determined by additive genetics. These estimates and comparisons are presented graphically in *Figure 1C*, and tabulated with confidence intervals and *p*-values in *Supplementary files 2* and *3*.

In the absence of a permutation scheme which is invariant under the null hypothesis of equal genetic and shared environmental factors, we additionally estimate $h^2 - c^2$ with bootstrapped confidence intervals to investigate the relative importance of these two components. In the fMRI data and in beta-band oscillations, where we also find significant genetic influences, we identify a differential effect of heritability and twins' shared environment (estimates with 95% confidence intervals, $h^2 - c^2 = 0.15\,[0.13, 0.16]$ for the fMRI and $h^2 - c^2 = 0.13\,[0.04, 0.22]$ for the beta-band results). Permutation tests for a significant contribution of heritability on each individual functional connection were performed, but no edges were significant after correcting for family-wise error at $\alpha = 0.05$. Parameter estimates for individual fMRI connections are shown in *Figure 1—figure supplement 1*.

To investigate whether our results were particular to our choice of parcellation, we re-ran our heritability analyses using the 15-dimensional ICA decomposition of resting-state fMRI recordings

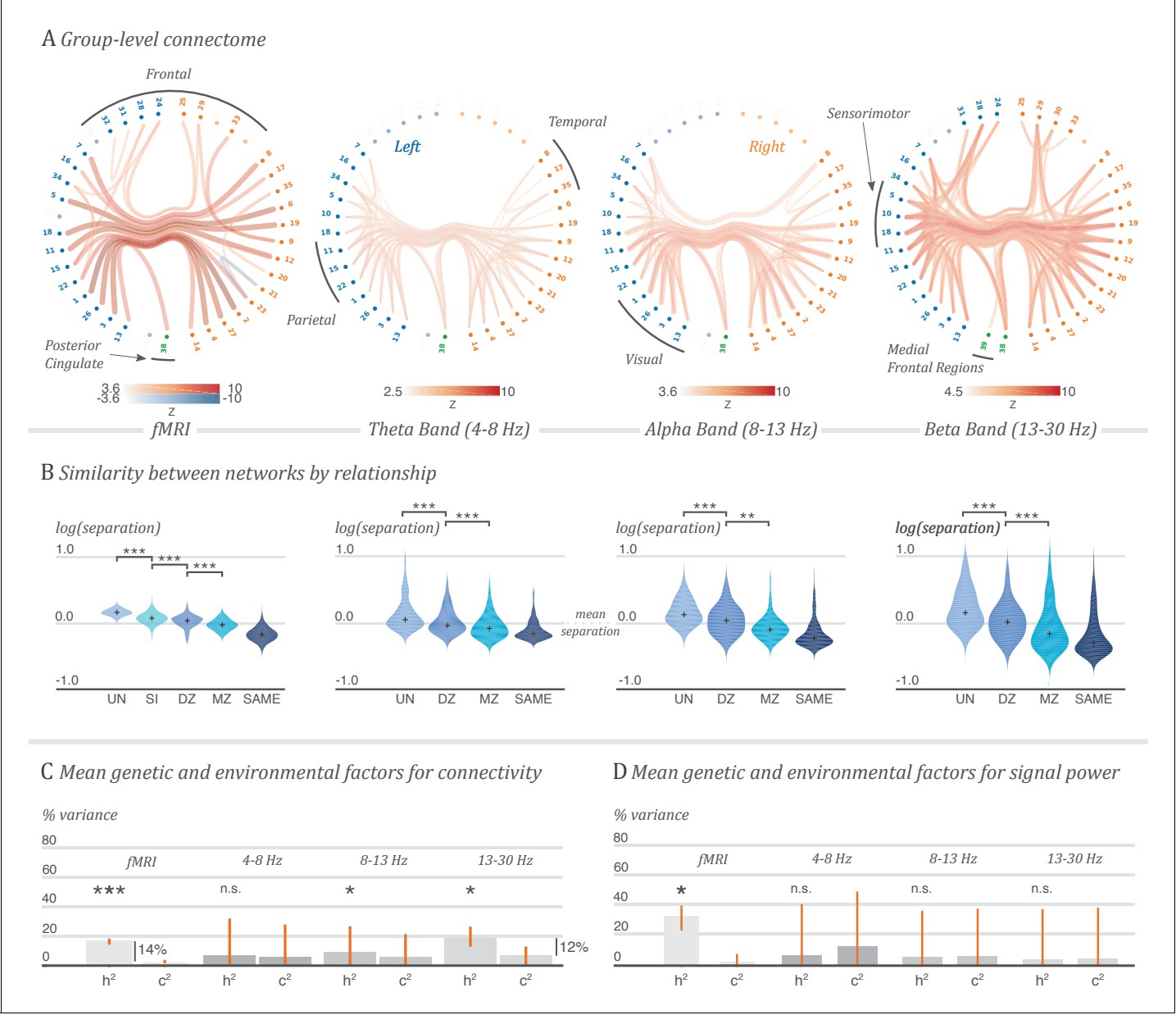

**Figure 1.** Contribution of genetic factors to functional connectivity outweighs that of the environment shared between twins. A. Grand average functional connectome for fMRI and for the theta, alpha and beta MEG oscillatory bands. The coloured edge maps show group-average network matrices for correlations in oscillatory amplitude in each band, and partial correlations in BOLD response, thresholded for visualisation. Nodes are annotated by cortical region, and labelled in *Supplementary file 1*. 3D renderings of these connectomes are shown in *Figure 1—videos 1 to 4*. B. Similarity of pairs of network matrices, separated by the relationship status of each pair. Subjects with a shared environment, and a greater proportion of shared genetics, have more similar organisation of neuronal coupling. Violin plots show distributions of distance values between pairs of network matrices estimated from single resting-state recording sessions, on a logarithmic scale relative to the mean network separation over all pairings, for pairs of unrelated subjects (UN), siblings (SI), dizygotic twin pairs (DZ), monozygotic twin pairs (MZ) and for repeated sessions with the same subject (SAME). C. Average genetic and shared environmental contributions to the variability of functional connectivity. The bar charts show the mean genetic component (heritability, $h^2$) and mean shared environment component ($c^2$) from a variance decomposition model fitted on each network edge, with 95% bootstrapped confidence intervals on the mean. Annotations indicate the difference in contribution between genes and the shared or developmental environment. The values of $h^2$ and $c^2$ are expressed as proportions of the total variance. The difference between their total and unity is $e^2$, the remaining environmental and measurement noise component. D. Average genetic and shared environmental contributions to the variability of oscillatory power, or BOLD response, in each ROI. Stars indicate significant differences in mean value for each of the displayed comparisons: ***p<0.001; **p<0.01; *p<0.05; n.s., not significant. The non-parametric *p*-values were computed by permutation and corrected for multiple comparisons over the 21 tests performed.

*Figure 1 continued on next page*

*Figure 1 continued*

DOI: https://doi.org/10.7554/eLife.20178.002

The following video and figure supplements are available for figure 1:

**Figure supplement 1.** Heritabilities of fMRI connection strengths.

DOI: https://doi.org/10.7554/eLife.20178.003

**Figure supplement 2.** Heritability of cortical curvature, and spatial profile of the heritability of functional connections.

DOI: https://doi.org/10.7554/eLife.20178.004

**Figure 1—video 1.** Animated rendering of the fMRI grand-mean network matrices shown in *Figure 1A*.

DOI: https://doi.org/10.7554/eLife.20178.005

**Figure 1—video 2.** Animated rendering of the theta band (4–8 Hz) grand-mean network matrices shown in *Figure 1A*.

DOI: https://doi.org/10.7554/eLife.20178.006

**Figure 1—video 3.** Animated rendering of the alpha band (8–13 Hz) grand-mean network matrices shown in *Figure 1A*.

DOI: https://doi.org/10.7554/eLife.20178.007

**Figure 1—video 4.** Animated rendering of the beta band (13–30 Hz) grand-mean network matrices shown in *Figure 1A*.

DOI: https://doi.org/10.7554/eLife.20178.008

released by the HCP. This parcellation contains entire resting-state networks as each node of the decomposition, yielding a functional connectivity matrix that describes inter-network, rather than intra-network relationships. We found significant genetic influences in the fMRI data ($h^2 = 0.29\,[0.26, 0.33]$, $p = 1 \times 10^{-4}$, uncorrected), but not for any of the activity recorded with MEG (uncorrected permutation-based $p$-values for $h^2 > 0$ were 0.69, 0.06 and 0.14 for the theta, alpha and beta bands respectively). The parameter estimates and confidence intervals are presented in *Supplementary file 4*.

The three-component model was also fitted to the variability in the logarithm of the signal power in each ROI, in both the fMRI and MEG data. Significant influences of genetic factors, on average over the nodes, were found only in the fMRI data (*Figure 1D*).

Finally, we fitted the heritability model to the cortical surface curvature of each subject, computing average $h^2$ within each ROI (*Figure 1—figure supplement 2*). Cortical curvature exhibits the highest heritability around the sensorimotor areas and insula (as has been reported previously, in an analysis using a different sample of the HCP subjects; *Sotiropoulos et al., 2015*). We compared this spatial profile of the heritability of curvature to the average heritability of network connections for each ROI within each modality (that is, for each ROI, the average $h^2$ of all connections involved with that ROI). In general, no significant positive correlations were found between the spatial arrangement of cortical curvature heritability and the heritability of network connections in particular ROIs. The exception was moderate correlation ($\rho = 0.39$) with the MEG theta band connectivity pattern, for which there was no significant heritability of connection strength in the first instance. (These data are presented in *Supplementary file 5*.)

## Discussion

Using resting-state fMRI and MEG recordings released as part of the HCP, we have constructed the functional network structure that expresses the coupling in MEG oscillatory power within three frequency bands, and fMRI partial correlation networks, among 39 ROIs. Based on these analyses, we make two key claims. First, genetic factors help to determine the strength and form of cortical oscillatory communication and functional connectivity. We have shown that the entire functional network structure is found to be more similar for two subjects the more closely they are related, and we estimate that the average heritability of individual connection strengths is about 15–18% for BOLD correlations, and between 13% and 26% (1% and 25%) for correlations in beta-band (alpha-band) power fluctuations. Second, genetic make-up is more important, on average, than the shared environment among twins when determining the strength of these couplings (with a difference of 13–16 percentage points in fMRI and 4–22% in the MEG beta-band).

Our results are drawn from two imaging modalities, fMRI and MEG, with functional connectivities estimated from the same set of ROIs. Our analyses with these technologies produce complementary assessments of functional connectivity: slow time scale couplings of functional activation indexed by BOLD response, and faster co-ordinations in oscillatory amplitude measured with MEG. (MEG is a

more direct measure of neuronal oscillatory activity, unaffected by vascular confounds.) We found, over both modalities (although, for the MEG results, only in the beta band), the same differential pattern of influence on functional connectivity from additive genetic factors and twins' shared developmental environments. Given the range of connectivity structures that are expressed in beta-band oscillations and in the slow time scale couplings measured in fMRI, these results suggest that genes have broad control over functional connectivity in the cortex, with a contribution that outweighs shared environmental factors.

These complementary results in fMRI and MEG alpha- and beta-band oscillations, drawn from the same parcellation, provide strong new support for a neural basis of the genetic influences on BOLD connectivities. The heritability of functional connectivity measured with electrophysiology has been reported before (*Posthuma et al., 2005*; *Schutte et al., 2013*; recording heritabilities of 20–75%, predominantly in the alpha and beta bands). However, these analyses were performed between EEG sensors, using synchronisation likelihood for network estimation, a measure that is sensitive to volume conduction artefacts. We use a connectivity metric that explicitly suppresses these artefacts (*Colclough et al., 2015*; *Brookes et al., 2012*; *Hipp et al., 2012*; *O'Neill et al., 2015*), reducing the influence of heritable anatomical features and cortical folding patterns on our results. Additionally, by working with cortical reconstructions of the oscillatory sources, we gain not only substantial artefact rejection (*Schoffelen and Gross, 2009*; *Hillebrand et al., 2005*) but also interpretability; the functional networks that we identify in the beta band and which drive our genetic analyses are mostly in motor and posterior regions, in correspondence with previous findings (*Hipp et al., 2012*; *Baker et al., 2014*; *Hillebrand et al., 2012*; *Brookes et al., 2011*; *Brookes et al., 2012*; *Mantini et al., 2007*; *Marzetti et al., 2013*; *de Pasquale et al., 2012*; *de Pasquale et al., 2016*). Our results therefore advance the strength of the evidence for a neural mechanism that mediates the genetic control over functional connectivity. Current data cannot, however, reveal the mechanistic details of this control. We can speculate that genes will control both network dynamics (in terms of synaptic strengths or conductance delays) as well as biophysical properties (such as the distribution of local neuron populations and their immediate feedback systems)—but in the future it may be possible to combine neuroimaging results with large-scale biophysical models (*Deco et al., 2008*; *Cabral et al., 2014*) to better understand these influences.

We did not find significant genetic control of functional connectivity in the theta frequency band. This may be in part because of the difficulty in cleanly estimating functional connectivity in MEG, particularly outside the alpha and beta bands (*Colclough et al., 2016*), but may also be because the 89 HCP subjects with resting-state MEG scans present a much smaller sample for study than the rfMRI HCP dataset. Even the conclusions from our alpha- and beta-band analyses, while coincident with the results in fMRI, are necessarily summary characterisations of the role of genes and environment on cortical oscillatory coupling. Our claims are based on significant results from robust non-parametric tests, but the confidence intervals on our parameter estimates are inevitably broad.

The primary parcellation that we employed consists of focal, contiguous regions. Our heritability analyses therefore reflect the genetic and shared environmental influences on the strengths of connections *between* the component nodes of the established resting-state networks (such as the default mode, motor, visual and dorsal attention networks). In an additional analysis, we found significant effects of additive genetic factors on the strength of *inter*-network connections in fMRI, using an alternative lower-dimensional parcellation based on 15 entire networks as nodes for the connectome. This suggests that genetic control of functional connectivity extends across multiple spatial scales of synchronisation. For oscillatory activity measured with MEG, it is not clear that it is meaningful to summarise, in the same manner as for an fMRI inter-network analysis, the collective behaviour of these gross network structures using a single time course: there is growing evidence from MEG that the extended networks from the fMRI literature are composed of smaller sub-networks that synchronise and de-synchronise on relatively fast time scales (*Baker et al., 2014*; *de Pasquale et al., 2012*; *de Pasquale et al., 2016*). That we found no significant genetic effects on the connections among this additional set of 15 networks in our MEG data may simply reflect the unsuitability of static whole-network parcellations for this modality.

We have taken a comprehensive approach to our heritability analyses by controlling for a wide range of potential confounds. Of the heritable effects of physiology, anatomy and noise on functional connectivity, the influences of signal power, cortical folding patterns and subject motion are perhaps most worthy of discussion.

Higher coupling strengths are commonly observed between regions with higher signal power in both fMRI and MEG estimates of functional connectivity. This effect can be caused just by the increased signal to noise ratio in the observed regions, rather than any difference in underlying coupling strength (*Friston, 2011*). We fitted a genetic and environmental factors model to the power in each ROI of our analysis. In our MEG data, we found no influence of these factors on the measured signal strength, and so can exclude this signal-to-noise effect as a likely confound in the beta-band MEG results. There was a significantly heritable component to signal power in the fMRI data—but this is not surprising in and of itself. To control for its influence on our results, we have included measures of the signal power in each node as confound regressors in all of our genetic analyses of functional connectivity. (Signal power is not a perfect proxy for the signal to noise ratio, as the measured power will be affected by fluctuations in both the noise and the signal.)

Cortical folding patterns have been shown to have modest heritability (*Sotiropoulos et al., 2015*; *Botteron et al., 2008*; *van Essen et al., 2014*); although the strength of genetic control and the mechanism of influence are not yet certain (*Tallinen et al., 2016*; *Gómez-Robles et al., 2015*; *Ronan and Fletcher, 2015*). This has particular implications for the MEG measurements, where the arrangement of cortical folds will have a strong impact on the signals measured outside the scalp, and on the leakage (or volume conduction) of signals between different sensors. Our connectivity estimates use methods that are robust to source leakage artefacts (*Colclough et al., 2015*; *Brookes et al., 2012*; *Hipp et al., 2012*; *O'Neill et al., 2015*), and the lack of strong correlations between the spatial profile of the heritability of cortical curvature and the heritability of edges associated with each ROI (as shown in *Figure 1—figure supplement 2* and *Supplementary file 5*) is good evidence, we believe, that this confound has not seriously impacted our findings.

Subjects' motion inside a scanner during resting-state recordings is heritable (*Couvy-Duchesne et al., 2014*), and motion is a known resting-state confound for functional connectivity analyses (*Siegel et al., 2016*). While our pre-processing steps include appropriate registration of images into static reference frames, and the removal of components from the data that are associated with motion, we have additionally included in our genetic analyses a summary measure of participants' movements within their fMRI scans, to reduce our co-measurement of heritable motion traits. Lastly, we are unable to specifically control for eye movement in our genetic analyses, as no tracked eye recordings are available for the HCP resting-state scans. (Independent components of the sensor data that correspond to eye movements are, however, removed during pre-processing.) However, it is worth noting that although saccades and micro-saccades are thought to be linked to gamma-band oscillatory patterns (*Muthukumaraswamy, 2013*; *Orekhova et al., 2015*), which we did not study, there is no strong evidence-base that they influence resting-state recordings at lower frequencies.

We used variance-component models to represent the observed variability in functional connectivity as a set of contributions from additive genetic factors and from the shared developmental environment in twin pairs. Our conclusions rest on a number of assumptions implicit in this model (*Boomsma et al., 2002*). These include an assumption that people choose their partners randomly; that the relevant genetic mechanisms are additive; and that there is no significant interaction between genes and the shared environment—an effect which has been reported to exist in various aspects of cognition (*Nisbett et al., 2012*). Most importantly, we assume that monozygotic and dizygotic twin pairs will equally share exposure to environmental factors in their upbringing. If this is an accurate assumption, then estimates of $c^2$ and $h^2$ are informative about the relative roles of developmental environment and additive genetics in the variation of functional connectivity phenotypes. While there is evidence in support of the equal-environment assumption (*Felson, 2014*; *Conley et al., 2013*), it has been challenged on the basis that monozygotic twins look more alike, behave more alike, and are treated more similarly than dizygotic twins (*Joseph, 1998*). Without entering into the discussion on this point, we note that if this assumption fails in our sample, it would lead to an overestimation of genetic heritability and an underestimation of the impact of developmental environment, therefore potentially weakening our conclusions on the differential effect of these two influences. Lastly, the $c^2$ term in our model cannot be solely identified with developmental environment. It encompasses all effects which are common between twins, which may include intra-uterine and mitochondrial influences, in addition to the shared environmental factors.

Our results add substantially to the evidence for significant genetic control of human cortical connectivity. Structural connectivity, the distribution of the white matter tracts across the brain that

enable communication, is known to be highly heritable (*Zhu et al., 2015*; *Jahanshad et al., 2012*; *Kochunov et al., 2015*). (A study of the heritability of structural connections with the HCP dataset estimated a mean heritability of 25%; *Sotiropoulos et al., 2015*) Previous studies of the heritability of functional connectivity with fMRI (which, in general, fail to control for subject motion, node power or brain size, and some of which focus instead on network topology), do also find significant genetic effects with heritability estimates lying between 20% and 60% (*Smit et al., 2008*; *Fornito et al., 2011*; *van den Heuvel et al., 2013*; *Thompson et al., 2013*; *Jansen et al., 2015*; *Glahn et al., 2010*). However, connectivity estimation is noisy, and as heritability is expressed as a fraction of the observed phenotypic variance (including measurement noise), we can expect the parameter estimates of heritability to change as network estimation methods improve. The important comparison, therefore, is of the relative importance of genes to the developmental and environmental factors shared by twins. Our observation of the stronger influence of the former over the latter on functional connectivity is a relationship shared in several other cognitive, behavioural and physiological traits measured in adults (*Polderman et al., 2015*).

Twin studies cannot probe the precise genetic mechanisms of these influences. Recent work (*Hawrylycz et al., 2015*; *IMAGEN consortium et al., 2015*) has correlated the profiles of gene expression in different regions of cortex, and compared these patterns of gene transcription to resting-state fMRI connectivity profiles, identifying sets of genes with co-expression patterns that reflect networks of functional connectivity. However, detailed investigations into the specific genes affecting healthy resting-state connectivity must await large datasets with coincident resting-state functional imaging and genotyping. (Genetic data will soon be available both for the HCP and for the UK Biobank imaging data.)

Taken together, our results provide strong additional evidence for a neural basis of the heritability of functional connectivity in the human brain. We identify genetic influences both in fMRI datasets and in (alpha- and beta-band) electrophysiological recordings, using functional network analyses on the same cortical parcellation. Our results more comprehensively control for a wide range of important confounds than previous work in this area. We have made particular emphasis of the increased importance of genetics over the developmental environment in determining cortical functional connectivity. The relevance and implications of this finding are widespread. Functional connectivity profiles are associated with intelligence, which is well-known to be heritable (*Bartels et al., 2002*; *Neisser et al., 1996*). But connectivity is also implicated with a very broad range of behavioural and life-style factors, including earning power, measures of health, various assessments of cognitive performance, and self-reported life satisfaction (*Smith et al., 2015*). As the findings of the Moving To Opportunity experiment in the United States make clear (*Ludwig et al., 2013*; *Chetty et al., 2015*), together with results from studies on the effects of schooling, socio-economic environment and adoption (*Nisbett et al., 2012*), the environment in which individuals develop can still make a significant impact on their performance and long-term outcomes. However, while genes are not the only important factor, results such as these presented here suggest that nature, more so than nurture, is a major force in determining integrated cognitive function and, by extension, cognitive capability.

## Materials and methods

### Subjects

Four 15-minute resting-state fMRI recordings from 820 subjects were collected as part of the HCP900 data release by the Human Connectome Project (*WU-Minn HCP Consortium et al., 2013*; *WU-Minn HCP Consortium et al., 2013*). Additionally, 89 of these subjects provided three 6-minute resting-state MEG recordings. All subjects are young adults (22–35 years of age) and healthy. Of the fMRI (MEG) subjects, there are 103 (19) monozygotic and 54 (13) dizygotic complete twin pairs. Zygosity was determined from subjects' genotypes when available, and otherwise by self-report.

HCP data were acquired using protocols approved by the Washington University institutional review board. Informed consent was obtained from subjects. Anonymised data are publicly available from ConnectomeDB (db.humanconnectome.org; *Hodge et al., 2016*). Certain parts of the dataset used in this study, such as the family structures of the subjects, are available subject to restricted data usage terms, requiring researchers to ensure that the anonymity of subjects is protected (*WU-Minn HCP Consortium et al., 2013*).

## fMRI analysis

Resting-state fMRI data were acquired with 2 mm isotropic spatial resolution and a temporal resolution of 0.72 s. The HCP provides comprehensively pre-processed data (*WU-Minn HCP Consortium et al., 2013*) that are mapped to a standard cortical surface using a multi-modal registration algorithm, MSMAll (*Robinson et al., 2014*; *Glasser et al., 2016*), and for which structured artefacts (with origins including motion, heartbeat and cerebro-spinal fluid) have been removed by a combination of ICA and FIX (*Salimi-Khorshidi et al., 2014*), FSL's automated noise component classifier.

We estimated functional connectivity between 39 fMRI-derived cortical ROIs. We used the parcellation employed in Colclough et al. (*Colclough et al., 2015*), which contains contiguous regions identified from a resting-state 100-dimensional group-ICA decomposition of fMRI data from the first 200 subjects of the HCP project. Most ROIs have a symmetric counterpart in the opposite hemisphere, except those on the midline such as the posterior cingulate cortex. The parcellation is shown in *Figure 2A*. This parcellation was chosen because the use of contiguous, focal nodes creates a comprehensible network model between individual regions. The low dimensionality of MEG data restricts us to parcellations of about this number of ROIs or less: as there are only a few hundred sensors, reconstructing time courses for more than approximately 60 different regions would be noisy or nonsensical. This upper bound is supported by models that categorise signals as originating within or without the MEG dewar, which suggest a limit of about 60 measurable cortical sources (*Taulu et al., 2005*), and by recently developed data-driven parcellations that only find around 70 unique identifiable parcels using combined MEG and EEG data (*Farahibozorg et al., 2017*). To investigate whether our results were particular to our choice of parcellation, we additionally ran our heritability analyses on a lower-dimensional decomposition: the 15-dimensional group-ICA map computed from the HCP900 data. The nodes in this parcellation represent entire (non-contiguous) functional networks (such as the default mode or motor networks), and the connectivity matrix therefore represents inter-network dependencies. The parcellation is shown in *Figure 2B*.

A single BOLD time course to represent each ROI was constructed using multiple spatial regression in FSLnets. Partial correlation matrices were constructed using the tools in FSLnets, using mild Tikhonov regularisation ($\lambda = 0.01$) and a conversion to Z-values with Fisher's transform, referenced to

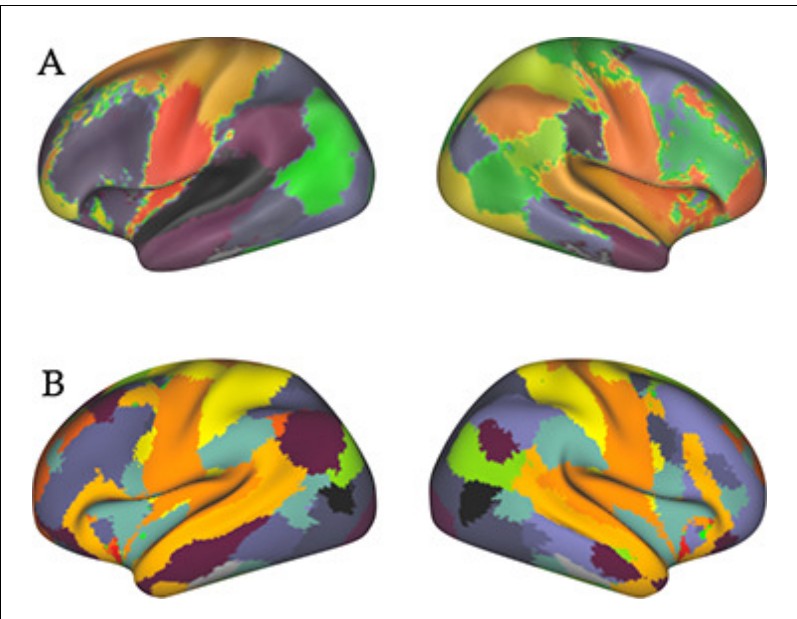

**Figure 2.** Regions of interest used for functional connectivity estimation. (A) Primary parcellation of 39 contiguous clusters, identified from a resting-state 100-dimensional group-ICA decomposition of fMRI data from the first 200 subjects of the HCP project. (B) The 15-dimensional fMRI ICA parcellation computed by the HCP as part of the S900 data release.
DOI: https://doi.org/10.7554/eLife.20178.009

the standard deviation of correlations of null data. Group-level networks were estimated as the mean Z-transformed correlation matrices over all sessions.

## MEG analysis

Resting-state MEG data were acquired on a whole-head Magnes 3600 scanner (4D Neuroimaging, San Diego, CA, USA). The data have been pre-processed to compensate for head movement, to remove artefactual segments of time from the recordings (which might relate to head or eye movement), identify recording channels which are faulty, and to regress out artefacts with clear temporal signatures (such as eye-blinks, saccades, muscle artefacts or cardiac interference) using ICA (*WU-Minn HCP Consortium et al., 2013*). Sensor-space data were down-sampled from 509 Hz to 300 Hz, with the application of an anti-aliasing filter.

MEG data from each session were source-reconstructed using a scalar beamformer (*Van Veen et al., 1997*; *Robinson and Vrba, 1999*; *Woolrich et al., 2011*). Pre-computed single-shell source models are provided by the HCP at multiple resolutions, registered into the standard co-ordinate space of the Montreal Neuroimaging Institute (MNI). Data were filtered into the 1–30 Hz band before being beamformed onto a 6 mm grid using normalised lead fields. Covariance estimation was regularised using PCA rank reduction. The rank was conservatively reduced by five more than the number of ICA components removed during preprocessing. Source estimates were normalised by the power of the projected sensor noise. Source-space data were filtered into theta (4–8 Hz), alpha (8–13 Hz) and beta (13–30 Hz) bands.

The same parcellations were employed for the MEG analysis as for the fMRI. As the MEG source reconstruction was performed over a volumetric grid in the MNI's standard space, rather than on the cortical surface, we used the volumetric versions of the fMRI ICA decompositions to form the MEG parcellations. A single time-course was constructed to represent each node as the first principal component of the ROI, after weighting the PCA over voxels by the strength of the ICA spatial map. This analysis yielded 39 time-courses for each frequency band and session for our principal parcellation (and 15 for the second).

One major confound when estimating connectivity in source-localised MEG is the spatially local bleeding of estimated sources from their true location into neighbouring regions. We compensate for these spatial leakage confounds, which can induce spurious connectivity estimates, using a symmetric orthogonalisation procedure (*Colclough et al., 2015*) to remove all shared signal at zero lag between the network nodes. This procedure is a multivariate extension of the orthogonalisation principle proposed in *Brookes et al., 2012*; *Hipp et al., 2012*; and *O'Neill et al., 2015*. It identifies the set of ROI time courses least displaced from the initial, uncorrected set, while enforcing mutual orthogonality between them, with no bias related to any reordering of the nodes. This approach suppresses any artificial correlations induced by spatial leakage, at the expense of any zero-lag connections of true neuronal origin. Lastly, power envelopes of the leakage-corrected ROI time-courses were computed by taking the absolute value of the Hilbert transform of the signals, low-pass filtering with a cut-off of 1 Hz, and down-sampling to 2 Hz (*Luckhoo et al., 2012*). We illustrate this stage in *Figure 3*.

Correlations between the power time-courses in each band were computed, converted to Z-values using Fisher's transform, and standardised with reference to the standard deviation of an empirical null distribution of correlations generated from time-courses with the same temporal properties as the data under test (see *Colclough et al., 2015*, for a detailed description). As for the fMRI networks, group-level networks were estimated as the mean Z-transformed correlation matrices over all sessions.

We use slightly different network estimation methods in the two modalities. Both are correlation measures, to maintain the similarity in analysis between the fMRI and MEG data. Partial correlation, used in the fMRI data, is one of the most accurate and robust measures in this modality (*Smith et al., 2011*; *Smith et al., 2013*; *Marrelec et al., 2006*; *Varoquaux and Craddock, 2013*). However, partial correlation can be difficult to estimate, even with regularisation, and it would not necessarily be the optimal choice for many smaller or lower quality fMRI datasets. An assessment of the repeatability of the most common network estimation measures in MEG showed that regularised partial correlation is relatively unreliable in this modality, and that the full correlation of the power envelopes of oscillatory activity (as we use here) was the most repeatable approach (*Colclough et al., 2016*). We have therefore attempted to use the best-practice inference methods

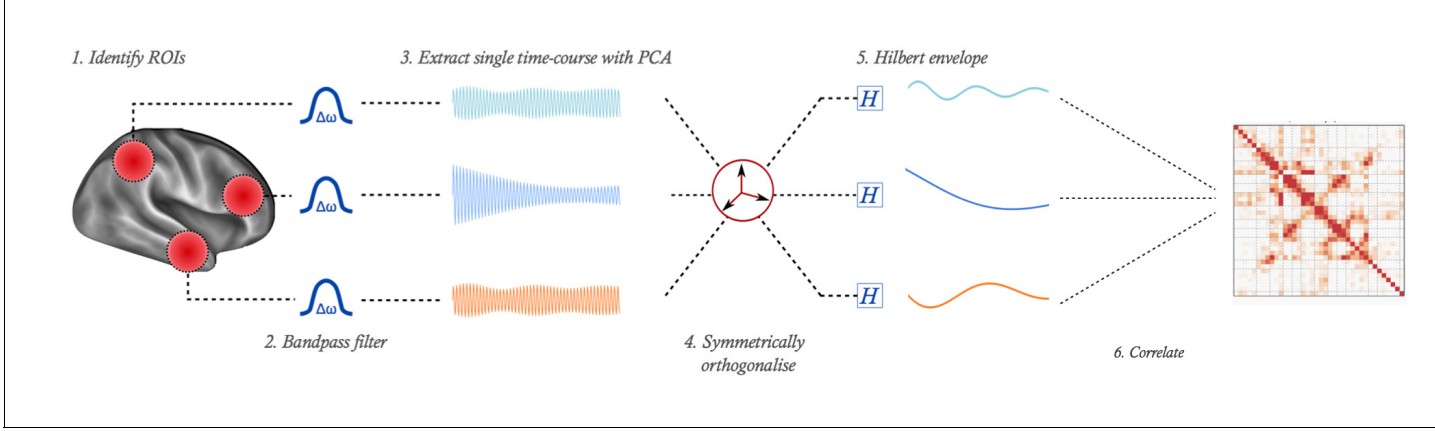

**Figure 3.** Illustration of pipeline for MEG functional connectivity estimation. From a whole-brain source-reconstruction, a single time-course is extracted to represent each ROI. These time-courses are bandpass filtered, then orthogonalised to remove shared signal which is potentially attributable to spatial leakage effects. The power envelope of each time-course is computed, then correlated to form the network matrix.
DOI: https://doi.org/10.7554/eLife.20178.010

in each modality. We did repeat our fMRI analysis using full correlation, rather than partial correlations, and found qualitatively similar results, with no difference in the significance of the claims we would make.

## Comparison of entire network structure

Similarity between pairs of functional networks was measured as the inverse Euclidean distance between the correlation matrices, after a logarithmic projection onto a Euclidean plane locally tangent to the Riemann manifold of positive semi-definite matrices (*Barachant et al., 2013*; *Ng et al., 2014*). This process can decouple the inter-relations between the elements within each positive-definite network matrix, and can improve the performance of classification algorithms when network matrices are used as discriminative features. It also provides for the definition of a true distance metric, where more similar matrices are projected to more proximate locations on the plane. A suitable Euclidean space for each modality and frequency band was found as the Euclidean plane tangent to the cone of positive semi-definite matrices, taking the geometric mean of the networks computed from all sessions as the tangent point. The separation between each pair of network matrices in each band was computed, forming different distributions of network similarity for pairs of subjects, split by the shared genetics of each pair. To assess whether shared environmental factors and shared genetics were associated with global network structure, we tested the difference in mean between the distributions of the logarithm of network separation for pairs of unrelated subjects and pairs of dizygotic twins, and between pairs of dizygotic twins and monozygotic twins, using a non-parametric *t*-test based on 20,000 permutations of the group labels. In the fMRI data, we were also able to compare unrelated subjects to siblings, and siblings to dizygotic twins.

## Three-component variance models

An ACE model for the subject-to-subject variability in connection strength was fitted for each network edge. This model splits the observed phenotypic variability into three factors: (A), additive genetics or *heritability* ($h^2$), (C), common environment ($c^2$), and (E), measurement error or external sources of variability. The three factors $h^2$, $c^2$ and $e^2$ are proportions of the total variance, and therefore normalised such that they sum to one. The APACE system of permutation inference (*Chen, 2014*) was employed, using the mean network matrix (over resting-state recording sessions) for each subject.

On each edge, we regressed out the effect of age, the square of age, sex, an age and sex interaction, an interaction between sex and the square of age, the cube root of intra-cranial volume and of cortical volume (both estimated with FreeSurfer), a measure of subject motion in the scanner (fMRI_-motion) and, for the fMRI data, the MR image reconstruction software version. Summary motion

estimates are only available for the fMRI recordings, but we use these values as a proxy measure of subjects' movements in both modalities. On each network edge, we included a regressor to account for heritable changes in node power, computed as the geometric mean of the power in each of the two nodes that each connection joins. We computed power simply as the standard deviation of the signal (fMRI time course or MEG power envelope) in each node. In the MEG data, where the standard deviation of the power of a virtual sensor is often correlated with its mean, we additionally include, in the same manner, a regressor formed from the ratio of the standard deviation to the mean of the power time course.

Finally, we also regressed out a measure of the noise passed by the beamformer for each subject. Our beamformer is described in *Woolrich et al. (2011)*, and the noise it passes at each voxel is the denominator of equation 4 in that paper,

$$\left( H^T(r_i) I \, / \, \sigma_e^2 H(r_i) \right),$$

where $H$ is the projection of the $N \times 3$ lead field matrix (for $N$ sensors) that maximises power at location $r_i$, and $\sigma_e^2 I$ is the noise covariance matrix. The scale of the noise is estimated from the smallest eigenvalue of the data covariance matrix. These voxel-wise noise estimates are scaled by the same weightings used to compute the ROI time courses, and averaged over ROIs to create a regressor estimating the noisiness of the beamformer for each subject.

Estimates of heritability were computed on each edge, with family-wise error corrected $p$-values computed by permutation, randomly shuffling monozygotic and dizygotic twin statuses 15,000 times. It is useful to consider a single, summary measure of functional connectivity for the entire functional networks. The mean values of $h^2$ and $c^2$ were computed over all functional connections; the $p$-value for mean heritability was computed by permutation, as above; confidence intervals for $h^2$, $c^2$ and $h^2 - c^2$ were computed by bootstrap re-sampling twin-pairs with replacement, 15,000 times.

The ACE model was also fitted to the logarithm of the variance of the corrected power envelopes (variance of the BOLD time course) in each ROI. This provided confidence that the conclusions of the ACE model for functional connectivity were not being driven by heritable effects in power or SNR within the network nodes. On each ROI, we removed the same set of regressors as for the functional connectivity analyses (save the power variables) before fitting the model.

Lastly, the ACE model was fitted within each ROI to estimates of cortical curvature for each of the subjects in the fMRI sample. We regressed out the effect of age, the square of age, sex, an age and sex interaction, the cube root of intra-cranial volume and of cortical volume, subject motion and the MR image reconstruction software version, before computing the mean heritability ($h^2$) over all points within each ROI. To compare spatial profiles of cortical folding with connectivity, we averaged the heritabilities of each network connection onto their constituent nodes, to create a spatial map suggesting the heritability of connectivity by ROI. These maps were correlated with the maps of heritability of cortical curvature in each ROI, and significance assessed by permuting ROIs 5000 times.

## Methodological notes

We applied a false discovery rate correction to the 21 principal statistical tests conducted for this paper to compensate for the multiple comparisons we perform (*Benjamini and Hochberg, 1995*). Uncorrected and corrected $p$-values are available in *Supplementary files 3* and *5*.

All analyses were performed in Matlab. MEG network analyses were performed with the MEG-nets software (github.com/OHBA-analysis/MEG-ROI-nets), fMRI network analyses with the FSL-nets software (fsl.fmrib.ox.ac.uk/fsl/fslwiki/FSLNets), and heritability analyses using APACE (the Advanced Permutation inference for ACE models (APACE) software is available at warwick.ac.uk/tenichols/apace).

## Acknowledgements

The authors would like to thank Matthew Brookes and Andrew Quinn for helpful discussions, and Paul McCarthy for his assistance in creating figures. Functional MRI and MEG data were provided by the Human Connectome Project, WU-Minn Consortium (Principal Investigators: David Van Essen and Kamil Ugurbil; 1U54MH091657) funded by the 16 NIH Institutes and Centers that support the NIH

Blueprint for Neuroscience Research; and by the McDonnell Center for Systems Neuroscience at Washington University. GLC is funded by the Research Councils UK Digital Economy Programme (EP/G036861/1, Centre for Doctoral Training in Healthcare Innovation); SMS by a Wellcome Trust Strategic Award (098369/Z/12/Z); TEN by the Wellcome Trust (100309/Z/12/Z) and the NIH (R01EB015611-01); AMW by the National Research Council of Brazil (CNPq, 211534/2013-7); SNS by the UK Engineering and Physical Sciences Research Council (EP/L023067); MFG by an NRSA fellowship (F30-MH097312, NIH); DCVE by the NIH (1U54MH091657); and MWW by the Wellcome Trust (106183/Z/14/Z) and the MRC UK MEG Partnership Grant (MR/K005464/1). This research was supported by the NIHR Oxford Biomedical Research Centre. The Wellcome Centre for Integrative Neuroimaging is supported by core funding from the Wellcome Trust (203139/Z/16/Z).

## Additional information

### Competing interests

David C Van Essen: Senior editor, *eLife*. The other authors declare that no competing interests exist.

### Funding

| Funder | Grant reference number | Author |
|---|---|---|
| Research Councils UK | Digital Economy Programme (EP/G036861/1, Centre for Doctoral Training in Healthcare Innovation) | Giles L Colclough |
| Medical Research Council | MRC UK MEG Partnership Grant (MR/K005464/1) | Giles L Colclough<br>Mark W Woolrich |
| Wellcome Trust | 098369/Z/12/Z | Giles L Colclough<br>Stephen M Smith<br>Thomas E Nichols<br>Anderson M Winkler<br>Stamatios N Sotiropoulos<br>Mark W Woolrich |
| Wellcome Trust | 100309/Z/12/Z | Giles L Colclough<br>Stephen M Smith<br>Thomas E Nichols<br>Anderson M Winkler<br>Stamatios N Sotiropoulos<br>Mark W Woolrich |
| Wellcome Trust | 106183/Z/14/Z | Giles L Colclough<br>Stephen M Smith<br>Thomas E Nichols<br>Anderson M Winkler<br>Stamatios N Sotiropoulos<br>Mark W Woolrich |
| Wellcome Trust | 203139/Z/16/Z | Giles L Colclough<br>Stephen M Smith<br>Thomas E Nichols<br>Anderson M Winkler<br>Stamatios N Sotiropoulos<br>Mark W Woolrich |
| National Institutes of Health | R01EB015611-01 | Thomas E Nichols<br>Matthew F Glasser<br>David C Van Essen |
| National Institutes of Health | NRSA fellowship (F30-MH097312) | Thomas E Nichols<br>Matthew F Glasser<br>David C Van Essen |
| National Institutes of Health | 1U54MH091657 | Thomas E Nichols<br>Matthew F Glasser<br>David C Van Essen |

| | | |
|---|---|---|
| Conselho Nacional de Desenvolvimento Científico e Tecnológico | CNPq,211534/2013-7 | Anderson M Winkler |
| Engineering and Physical Sciences Research Council | EP/L023067 | Stamatios N Sotiropoulos |
| National Institute for Health Research | NIHR Oxford Biomedical Research Centre | Mark W Woolrich |

The funders had no role in study design, data collection and interpretation, or the decision to submit the work for publication.

### Author contributions

Giles L Colclough, Conceptualization, Software, Formal analysis, Validation, Investigation, Visualization, Methodology, Writing—original draft, Project administration, Writing—review and editing; Stephen M Smith, Conceptualization, Data curation, Software, Supervision, Funding acquisition, Investigation, Visualization, Methodology, Project administration, Writing—review and editing; Thomas E Nichols, Stamatios N Sotiropoulos, Software, Investigation, Methodology, Writing—review and editing; Anderson M Winkler, Investigation, Methodology, Writing—review and editing; Matthew F Glasser, Data curation, Writing—review and editing; David C Van Essen, Data curation, Funding acquisition, Writing—review and editing; Mark W Woolrich, Conceptualization, Software, Supervision, Funding acquisition, Investigation, Methodology, Project administration, Writing—review and editing

### Author ORCIDs

Giles L Colclough http://orcid.org/0000-0003-1074-7186
David C Van Essen http://orcid.org/0000-0001-7044-4721

### Ethics

Human subjects: HCP data were acquired using protocols approved by the Washington University institutional review board. Informed consent was obtained from subjects. Anonymised data are publicly available from ConnectomeDB (db.humanconnectome.org; Hodge et al., 2016). Certain parts of the dataset used in this study, such as the family structures of the subjects, are available subject to restricted data usage terms, requiring researchers to ensure that the anonymity of subjects is protected (Van Essen et al., 2013).

### Decision letter and Author response

Decision letter https://doi.org/10.7554/eLife.20178.020
Author response https://doi.org/10.7554/eLife.20178.021

# Additional files

### Supplementary files

• Supplementary file 1. Index of ROI numbers.
DOI: https://doi.org/10.7554/eLife.20178.011

• Supplementary file 2. Parameter estimates and 95% confidence intervals for the mean genetic and shared environmental contributions to the observed phenotypic variability in functional connectivity and signal power, using the 39-dimensional parcellation derived from high-dimensional ICA on fMRI data.
DOI: https://doi.org/10.7554/eLife.20178.012

• Supplementary file 3. $p$-values for permutation-based significance tests performed for the strength of genetic factors, both before and after a false discovery rate correction for multiple comparisons over the 21 tests performed in this article.
DOI: https://doi.org/10.7554/eLife.20178.013

• Supplementary file 4. Parameter estimates and 95% confidence intervals for the mean genetic and shared environmental contributions to the observed phenotypic variability in functional connectivity and signal power, using the 15-dimensional ICA parcellation from fMRI data.
DOI: https://doi.org/10.7554/eLife.20178.014

• Supplementary file 5. Correlations over ROIs, with permutation-based $p$-values, between the average heritability of cortical curvature in each ROI and the average heritability of connections from each ROI. A lack of strong positive correlations suggests that any heritability in cortical curvature is not driving the heritability observed in functional connection strengths. $p$-values are given both uncorrected, and after a false discovery rate correction for multiple comparisons over the 21 tests performed in this article.
DOI: https://doi.org/10.7554/eLife.20178.015

### Major datasets

The following previously published datasets were used:

| Author(s) | Year | Dataset title | Dataset URL | Database, license, and accessibility information |
|---|---|---|---|---|
| David C Van Essen, Stephen M Smith, Deanna M Barch, Timothy E J Behrens, Essa Yacoub, Kamil Ugurbil, WU-Minn HCP Consortium | 2013 | The Human Connectome Project HCP900 data release | http://www.humanconnectome.org/study/hcp-young-adult/document/900-subjects-data-release | Open access dataset available from ConnectomeDB (https://db.humanconnectome.org/app/template/Login.vm). Account registration is required and access to certain data elements such as family structure is subject to restricted use terms (please see http://www.humanconnectome.org/study/hcp-young-adult/data-use-terms) |
| L J Larson-Prior, R Oostenveld, S Della Penna, G Michalareas, F Prior, A Babajani-Feremi, J-M Schoffelen, L Marzetti, F de Pasquale, F di Pompeo, J Stout, Mark W Woolrich, Q Luo, R Bucholz, P Fries, V Pizella, G Romani, M Corbetta, A Z Snyder, WU-Minn HCP Consortium | 2013 | The Human Connectome Project HCP900 MEG data release | http://www.humanconnectome.org/study/hcp-young-adult/document/900-subjects-data-release | Open access dataset available from ConnectomeDB (https://db.humanconnectome.org/app/template/Login.vm). Account registration is required and access to certain data elements such as family structure is subject to restricted use terms (please see http://www.humanconnectome.org/study/hcp-young-adult/data-use-terms) |

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
