## [Decision Letter]

Thank you for submitting your article "The heritability of multi-modal connectivity in human brain activity" for consideration by *eLife*. Your article has been reviewed by three peer reviewers, one of whom, Jack Gallant (Reviewer #1), is a member of our Board or Reviewing Editors and the evaluation has been overseen by Sabine Kastner as the Senior Editor. The following individuals involved in review of your submission have agreed to reveal their identity: David Glahn (Reviewer #2).

The reviewers have discussed the reviews with one another and the Reviewing Editor has drafted this decision to help you prepare a revised submission.

Although the Discussion highlighted some hesitation about the publication of this work in *eLife*, the reviewers have decided to ask for a revision, which should deal with (1) additional analyses, (2) novelty issues, (3) other smaller issues. Of particular concern is that we are not yet convinced that this work is sufficiently novel; the authors will have to make the case for that.

Summary:

This study examines genetic influences on resting state MRI and MEG connectivity. The article is clearly written and the data analysis procedures are reasonable as far as they go. However, the genetic influences on resting state have been reported several times using similar methods, and at this point it is unclear whether this paper is appropriate for publication in *eLife*, or rather whether it merely reflects an incremental increase in scientific knowledge. Furthermore, the current analyses appear to be insufficient. The MEG results are rather underpowered and many essential analytical controls are missing. After consultation the reviewers decided that the paper should be returned to the authors for revisions focusing on (1) elucidating the novel contributions of the work, (2) performing additional analysis as suggested and (3) dealing with a variety of smaller issues.

Essential revisions:

1) The genetic influences on resting state have been reported several times using similar methods. However, the current paper does not really include any detailed comparison of the current results with those reported previously, so it is difficult to judge the novelty of this contribution. In revision the authors should include a detailed discussion of precisely which aspects of the paper are novel. This may require further data analysis apart from what is requested as controls below. It would also be helpful if the authors would make more of an effort to explain some potential causal mechanisms that might underlie their reported relationships.

2) Additional analyses should be included to account for potential contamination of the FC data by confounding factors having nothing to do with cortical connectivity or communication, but which are nevertheless heritable. The obvious candidates here are body motion, head motion, eye movements, or other physiological factors.

3) The results are all based on one particular parcellation. Some evidence should be provided that the results do not depend on this particular choice. The optimal way to address this problem would be to rerun the analysis pipeline using several different parcellation schemes.

4) There appear to be several differences between the way that the fMRI and the MEG data were processed. These should be justified and explained, or a more consistent approach should be used.

Reviewer #1:

This is a fine study as far as it goes and it includes several good controls for potential contaminating factors (though these could be substantially improved). However, the paper is going to require revision and additional data analysis before it is suitable for publication.

Although the authors show THAT genetics influences FC, they provide no information about WHY genetics influences FC. Which of the many mechanisms that contribute to observed BOLD FC are influenced by genetics? As the authors note, several of the results reported here (such as the heritability of FC from EEG data) have been reported already in previous studies. It is claimed here that these previous studies are less interpretable than the current study. That may be true, but then the authors need to provide the interpretation. If not then this paper loses a lot of its potential novelty and impact and I am not sure that it will be suitable for publication in this venue. Some effort should be made to explain several of the more interesting and unusual effects as well. For example, it is stated that there is significant heritability in the alpha and beta bands of the MEG data but not in the theta band. Why? What is a plausible mechanism that would generate this pattern of results?

There is one major class of potential confounds which appear to be given short shrift here: potential contamination of the FC data by confounding factors having nothing to do with cortical connectivity or communication, but which are nevertheless heritable. The problem is that genetic factors might influence processes that are well known to influence correlations in MRI data (and likely MEG data as well). For example, if genetics causes some people to wiggle more in the magnet, that is obviously going to influence FC. (In fact I am fairly certain that I saw another study recently that reported just that effect, though unfortunately I am unable to find it now.)

To take just one specific example of the above concern, it is reported that the visual system has a high degree of FC in the MEG data. The most likely candidate here would be eye movements, which will clearly affect MEG correlations, and which may also be influenced by genetics. However, I didn't see any report here that the eyes were tracked properly, or that the eye tracking data were regressed out when calculating FC. It is precisely these sorts of potential confounds (i.e., genetic influences operating on behavior that in turn influences FC rather than operating on FC directly) that must be scrupulously accounted for before publication.

Perhaps one good way to get an intuitive handle on these sorts of potential indirect genetic influences would be to run separate analyses to determine the heritability of the various confounds that are known to affect FC, such as motion, eye movements, beamforming errors (in the MEG data). For that matter other factors that might affect FC indirectly might also be heritable. BOLD data taken directly off the scanner are non-Gaussian, and these are Gausianized during pre-processing by a necessarily imprecise procedure. Is the distribution of raw BOLD signals heritable? All these seem plausible, and all could potentially affect FC. A systematic analysis of these factors would seem to be critical for making any strong claims about direct genetic influences on FC.

Smaller issues:

The claims in the Abstract and Discussion are a bit over-stated in several places given the results because they do not make any reference to places where no heritability is found. For example, while effects in the MEG alpha and beta frequency bands are observed, this is not true for the theta frequency band.

The figures and the supplementary movies in this paper are rather poor, especially the connectivity figures. There are many excellent choices for visualization software these days.

Issues related to data analysis:

The pre-processing and data analysis procedures are necessarily complicated, and of course it is always possible that some decision that was made during those procedures might have biased the results. On the other hand, if we started requiring every single step to be addressed in multiple ways to ensure against this sort of bias then none of us would get anywhere! So in the next few paragraphs I only ask for further work on data analysis steps that I think could be particularly problematic, or where additional analysis might provide worthwhile enlightenment.

The entire data analysis procedure is based on the HCP ICA parcellation. It would inspire more confidence in the conclusions if some analysis was provided that showed whether the choice of parcellation scheme makes any difference on the results or the conclusions. I don't think that it would make much difference in the results if one compared the HCP ICA parcellation to, say, the Glasser HCP parcellation, because these are both fine-scale parcellations that are likely to include more spatial information than can really be supported by genetics data anyway. (In the case of the MEG data, the HCP ICA parcellation may very well have higher resolution than can be supported by either the genetics or the MEG.) It would be immensely helpful to know whether a coarser parcellation schemewould change the results, and I think that knowing the answer to this question might also help us address the question of why these apparent genetic influences are found.

It looks as if subject motion was removed during pre-processing in both the fMRI and the MEG data, but subject motion was still used as a regressor during ACE modeling in the fMRI data. Subject motion is a problem in both fMRI and MEG, so if it was a problem in the fMRI data even after pre-processing why was pre-processing sufficient for the MEG data? At a minimum it seems that it should also be included in the MEG modeling analysis.

Very small confusing issues:

This is a side point, but I find it surprising that it is useful/helpful to regress out the MR image reconstruction software version in ACE modeling. This suggests that these data really do live at the very limit of sensitivity, and that confounds and bias really can creep into the data even after pre-processing. It seems like the best practice would be to use the same version of the software to process all the data and I find it a bit worrisome that this was not done.

In Subsection “E. Three-component variance models” it is stated that the variance partitioning analysis addresses non-negative portions of the variance. Why do these procedures produce negative variance estimates at all? Is this just statistical error?

Reviewer #2:

Colclough and colleagues examine the genetic influences on resting state MRI and MEG connectivity in "The heritability of multi-modal connectivity in human brain activity." Subjects included 461 individuals with MRI data and a group of 61 primarily overlapping individuals with MEG data all from the Human Connectome Project. Subjects were from extended twin pairs or unrelated individuals. The goals of the article are to estimate the heritability and common genetic influences on resting state and MEG measures of connectivity. The article is clearly written and the analytic plan is both complex and reasonable. Unfortunately, the findings represent only an incremental increase in scientific knowledge, as the genetic influences on resting state have been reported several times using conceptually similar methods. While the MEG results are novel, that analysis is rather underpowered, as the authors appropriately note in the Discussion. Thus, while I believe the findings should be reported, the authors should either describe the work as replication or conduct additional analyses that will increase novelty.

Reviewer #3:

This is an excellent article, which combines state-or-the-art resting-state MEG analysis methods with new approaches to assessing heritability and environmental facts in this type of electrophysiological dataset. As such it both helps to drive the field forward and will be of general interest to the *eLife* readership. I recommend publication after the following issues are addressed:

1) I think the authors should clarify why 39 cortical regions are chosen, as opposed to other common parcellation schemes, which often have 50-100 regions.

2) I'm surprised that the Authors did not at least attempt to reconstruct networks in the higher gamma band. Why only theta/alpha/beta?

3) One of my main concerns is the use of partial correlations in the FMRI analysis and simple correlations in the MEG network analysis. Why this difference? The Authors cite two of their own papers as justification but this not particularly convincing. Could we have a short justification here?

4) The authors state that the beta-band "exhibits broad connectivity over the whole cortex". That doesn't really appear to be the case looking at Figure 1. In ay case, what does that statement actually mean – it's quite a woolly phrase that is not backed up by any kind of quantification.

5) The authors look at both genetic and environmental factors in terms of contribution to signal power (in both FMRI and MEG), partly as a control for these as confounds in the connectivity analyses, but also as interesting exploratory analyses in their own right. I have two concerns here. 1) Can the authors please describe how "power" was assessed in the FMRI signal? I assume this is some measure of temporal variation around the mean of the voxel time series? 2) For the MEG, the power passed by a beamformer in an RSN analysis can be quite dependent on geometric effects (even after weights normalisation). Wouldn't a better measure of 'activity" in the MEG amplitude envelopes be some measure of temporal variability? The standard deviation (SD) is often used, but the SD of the virtual-sensor and the Mean are often correlated in beamformer reconstructions, so a better measure might be some proportional change (e.g. SD/Mean). "Activity" in both the FMRI and MEG time series could thus be assessed using the same (or very similar) metric.

6) In the ACE model, quite a few nuisance covariates are regressed out of the model before heritability/environmental effects were assessed. How were these parameters (and their second-order versions) chosen? In addition, when regressing out some many parameters, is there an issue with statistical power in the MEG analyses as some of the groups only have relatively few subjects (i.e. 11 monozygotic twin pairs)?

7) Finally, with these "nuisance" regressors Is there a potential problem with interaction with heritability? For example age is 100% matched in the twin-pairs, but is presumably not matched for the non-related pairings.

---

## [Author Response]

Essential revisions:1) The genetic influences on resting state have been reported several times using similar methods. However, the current paper does not really include any detailed comparison of the current results with those reported previously, so it is difficult to judge the novelty of this contribution. In revision the authors should include a detailed discussion of precisely which aspects of the paper are novel. This may require further data analysis apart from what is requested as controls below. It would also be helpful if the authors would make more of an effort to explain some potential causal mechanisms that might underlie their reported relationships.

In summary, the novel contributions of this paper are:

a) Evidence not just of the heritability of functional connectivity, but that the genetic influences outweigh those of the shared developmental environment between twins.

b) The first reporting of the heritability of functional connectivity measured with

source-reconstructed electrophysiology.

c) Evidence that the genetic control of functional connectivity has an

electrophysiological basis, through the use of MEG functional connectivity analyses on the same cortical parcellation as the fMRI results. This analysis increases confidence that the genetic influence on functional connectivity is neuronal in origin, compared to previous EEG studies, by controlling for the heritable confounds associated with those approaches.

d) An analysis of the genetic and shared environmental influences of functional

connectivity that for the first time offers comprehensive control for a wide range of potential confounds.

In our revised submission, we have extended the comparison of our work to previous papers, and added discussion of potential causal mechanisms. Before highlighting these changes, we expand on these novel contributions above.

a) Estimates of heritability are highly contingent on the measuring apparatus. The noisier the measure of the phenotype, the lower the estimate of heritability, *h^2^*, because the three components of the analysis (genes, shared environment and noise) must sum to one. The important question, then, is what the sensible scale of comparison should be for heritability. Arguably, the most appropriate choice is *c^2^*, which captures (among some other effects) the environmental and developmental influences that are shared between twins. Investigating the relative importance of these two factors allows us to make important claims about the genetic control of functional connectivity, despite the noisiness of the measure. This is a perspective which until now has been absent from the literature, but which makes the results comparable to analyses of heritability in other fields relevant to investigations of functional connectivity, and of interest to a wide audience of the general public.

c) The evidence for genetic control of functional connectivity that we present from source-space MEG analyses is a critical advance on the evidence from sensor space EEG recordings. Sensor space connectivity estimation is a long way from the best practice that can be achieved in MEG (and potentially in high density EEG) because, in addition to the greater susceptibility of sensorspace data to artefacts, much of the connectivity may be driven purely by volume conduction effects, which are likely to be highly heritable, reflecting heritable anatomical and cortical folding traits. By comparison, functional connectivity in source space is much more interpretable as being neuronal in origin – both from the reduction in biological noise and the suppression of spatial leakage effects. Furthermore, the specific network edges that carry functional connectivity and drive the genetic effect can be inspected to see if they are in plausible brain areas. In our case, the alpha-band networks are in occipital areas, and the beta-band in motor and posterior regions, in line with previous findings (Hipp et al., 2012; Brookes et al., 2011, 2012; Baker et al., 2014; Mantini et al., 2007).

d) The reviewers had a strong focus on the need to account for a wide range of potentially heritable confound factors that are being captured within the measures of functional connectivity, in addition to the concerns over volume conduction or source leakage discussed above. We agree with the reviewers, and after extending our approach in line with their suggestions, we are confident that our heritability analysis controls for these confounds more comprehensively than any previous study, in any imaging modality. This represents a major strengthening of our understanding of the genetic control of functional connectivity.

We now highlight the specific changes we have made to the paper to address the requests of the reviewers in their concerns over novelty. This includes a detailed comparison of the current results with those reported previously; a discussion of which aspects of the paper are novel; and discussion of potential causal mechanisms.

First, we have expanded our Discussion with a new paragraph to be more explicit about the contribution of our electrophysiology results, and to provide a more detailed exploration of possible mechanisms for genetic control, together with a more explicit comparison to the existing literature, “These complementary results in fMRI and MEG alpha- and beta-band oscillations, drawn from the same parcellation, provide strong new support for a neural basis of the genetic influences on BOLD connectivities. […] We can speculate that genes will control both network dynamics (in terms of synaptic strengths or conductance delays) as well as biophysical properties (such as the distribution of local neuron populations and their immediate feedback systems)—but in the future it may be possible to combine neuroimaging results with large-scale biophysical models (Cabral et al., 2014; Deco et al., 2008) to understand these influences.”

We have altered later parts of the Discussion to contextualize our results, and to be more explicit about our contributions in comparison to the fMRI and diffusion MRI literature, “Our results add substantially to the evidence for significant genetic control on human cortical connectivity. Structural connectivity, […] Our observation of the stronger influence of the former over the latter on functional connectivity is a relationship shared in several other cognitive, behavioural and physiological traits measured in adults (Polderman et al., 2015).”

We conclude with an overview of the key contributions of the paper, “Taken together, our results provide strong additional evidence for a neural basis of the heritability of functional connectivity in the human brain. […] We have made particular emphasis of the increased importance of genetics over developmental environment in determining cortical functional connectivity. The relevance and implications of this finding are widespread”

2) Additional analyses should be included to account for potential contamination of the FC data by confounding factors having nothing to do with cortical connectivity or communication, but which are nevertheless heritable. The obvious candidates here are body motion, head motion, eye movements, or other physiological factors.

We agree with the reviewers that appropriate control of possible confounds to the heritability analysis is important, if we are to interpret the results as reflecting the genetic and environmental influences on functional connectivity alone.

Most of the literature we surveyed on the genetic influences of connectivity control for demographics in their genetic analyses (age and sex, for example). Some (for example, Fu et al., 2005, and Sinclair et al., 2015 – both fMRI studies) also control for participants’ motion in the scanners.

In our initial submission, we felt that these controls did not go far enough. On top of age and sex demographics, we included measures of brain volume, and of head motion (in fMRI). We were particularly concerned that our results might simply reflect heritable changes in anatomy that influence how our signal is measured (such as cortical folding patterns), or reflect heritable patterns in signal power (if, for example, higher connectivities are observed between two nodes simply because those nodes have higher SNRs). We therefore regressed out measures of node power from our analysis, and furthermore performed control analyses for the heritabilities of power and of cortical curvature. Finally, previous electrophysiological studies of the heritability of connectivity, which have tended to measure the synchronisation likelihood between sensors, will be sensitive to any heritable anatomical differences that create similar profiles of volume conduction. We instead used an approach for connectivity analysis in MEG that was defined in source-space and removed any confounding effects attributable to magnetic field spread (or source leakage).

The reviewers have suggested several additional measures that are important to account for. We have addressed their concerns to the extent to which it is possible within the scope of the available dataset, by now including the age^2^ * sex interaction, a measure of head motion, a measure of beamformer noise, and an additional measure of signal power in the MEG analyses. We have also expanded our discussion of the pre-processing pipelines used to remove structured noise and physiological artefacts from the data.

Eye movement data were unfortunately not collected as part of the HCP imaging protocol. We have expanded our Discussion accordingly, where we also note that independent components of the sensor data that correspond to eye movements are, however, removed during preprocessing. The paragraphs in our Discussion that focus on potential confounds now read, “We have taken a comprehensive approach to our heritability analyses by controlling for a wide range of potential confounds. Of the heritable effects of physiology, anatomy and noise on functional connectivity, the influences of signal power, cortical folding patterns and subject motion are perhaps most worthy of discussion.”

“Higher coupling strengths are commonly observed, in both fMRI and MEG estimates of functional connectivity, between regions with higher signal power. […] To control for its influence on our results, we have included measures of the signal power in each node as confound regressors in all of our genetic analyses of functional connectivity, although it is not a perfect proxy for the signal to noise ratio, as the measured power will be affected by fluctuations in both the noise and the signal.”

“Cortical folding patterns have been shown to have modest heritability (Botteron et al., 2008; Sotiropoulos et al., 2015; Van Essen et al., 2014); although the strength of genetic control and the mechanism of its influence are not yet certain (Gomez-Robles et al., 2015; Ronan & Fletcher, 2015; Tallinen et al., 2016). [...] Our connectivity estimates use methods that are robust to source leakage artefacts (Brookes et al., 2012; Colclough et al, 2015; Hipp et al., 2012; O’Neill et al., 2015), and the lack of strong correlations between the spatial profile of the heritability of cortical curvature and the heritability of edges associated with each ROI (as shown in supplementary figure 3 and supplementary table 5) is good evidence, we believe, that this confound has not seriously impacted our findings.”

“Subjects’ motion inside a scanner during resting-state recordings is heritable (Couvy- Duchesne et al., 2014), and motion is a known resting-state confound for functional connectivity analyses (Siegel et al., 2016). […] On the other hand, it is worth noting that although saccades and microsaccades are thought to be linked to gamma-band oscillatory patterns (Muthukumaraswamy, 2013; Orekhova et al., 2015), which we are not studying, there is no strong evidence-base that they influence resting-state recordings at lower frequencies.”

Our Material and methods section has been adjusted to read, “On each edge, we regressed out the effect of age, the square of age, sex, an age and sex interaction, an interaction between sex and the square of age, the cube root of intracranial volume and of cortical volume (both estimated with FreeSurfer), a measure of subject motion in the scanner (fMRI_motion) and, for the fMRI data, the MR image reconstruction software version. […] In the MEG data, where the standard deviation of the power of a virtual sensor is often correlated with its mean, we additionally include, in the same manner, a regressor formed from the ratio of the standard deviation to the mean of the power time course.”

“Finally, we also regressed out a measure of the noise passed by the beamformer for each subject. […] These voxel-wise noise estimates are scaled by the same weightings used to compute the ROI time courses, and averaged over ROIs to create a regressor estimating the ‘noisiness’ of the beamformer for each subject.”

3) The results are all based on one particular parcellation. Some evidence should be provided that the results do not depend on this particular choice. The optimal way to address this problem would be to rerun the analysis pipeline using several different parcellation schemes.

We have re-run our analysis using an additional parcellation, the 15-dimensional ICA decomposition from the HCP data. This choice was motivated in part by the limitations of MEG data, whose effective spatial resolution cannot support high-dimensional parcellations, and in part by the reviewers’ suggestions that we investigate the heritability of inter-network connections rather than intra-network connections. The 15-dimensional parcellation is well-suited for this, as it provides entire RSNs as the ‘nodes’ of the functional network analysis (including the DMN, DAN, motor network, visual networks). To our knowledge, such a parcellation has not previously been used in the literature for functional network analyses in MEG.

In the fMRI data, we find a higher mean heritability (29%) with the lower-dimensional parcellation than our original choice, and the difference of influence between genes and the environment shared between twins is also retained.

In the smaller MEG dataset, we find no significant heritability on average over the network connections, in any frequency band, using the 15-dimensional parcellation. This result is consistent with the fact that conventional (fMRI) RSNs appear only partially in resting MEG data, or are comprised of many sub-networks that synchronise at relatively fast timescales. This makes the use of this particular parcellation inappropriate for MEG. Building reliable and informative data-driven (e.g. lower-dimensional) parcellations in MEG is still very much an open, non-trivial research problem, and investigation of a low-dimensional parcellation that would yield positive results in MEG is beyond the scope of the current work.

In summary, we find the fMRI result encouraging, and suggest that the lack of a result in the MEG data highlights issues in finding appropriate parcellations for this modality.

We have included this paragraph in our Material and methods section, “To investigate whether our results were particular to our choice of parcellation, we additionally ran our analysis on a lower-dimensional decomposition: the 15-dimensional group-ICA map computed from the HCP900 data. The nodes in this parcellation represent entire functional networks (such as the default mode or motor networks), and the connectivity matrix therefore represents inter-network dependencies.”

Altered our Results section, “To investigate whether our results were particular to our choice of parcellation, we reran our heritability analyses using the 15-dimensional ICA decomposition of restingstate fMRI recordings released by the HCP..[…] The parameter estimates and confidence intervals are presented in Supplementary file 3.”

And added to our Discussion as follows, “The primary parcellation that we employed consists of focal, contiguous regions. […] That we found no significant genetic effects on the connections between this additional set of 15 networks in our MEG data may simply reflect the unsuitability of static whole-network parcellations for this modality.”

4) There appear to be several differences between the way that the fMRI and the MEG data were processed. These should be justified and explained, or a more consistent approach should be used.

The three reviewers have touched upon three differences in the way the fMRI and MEG data are processed: in the pre-processing steps, in the network estimation method, and in the correction for confounds within the heritability analyses.

We have discussed the improvements we have made to our confound controls in point 2) above, which now include using the same regressors in each modality, apart from those that are modality-specific.

The pre-processing steps are necessarily slightly different in detail between the two modalities, but generally take the same approach of: movement compensation – registration – removal of artefactual channels and epochs – ICA decomposition and removal of noise components – source reconstruction. The registration algorithms are obviously different between modalities, and the ICA noise removal process is based on an automated classification algorithm for the fMRI data only, because it has only been trained on these data. The pre-processing stages for each modality have all been performed by the HCP, and we believe it is fair to say that these pre-processing pipelines reflect the current state of the art in the pre-processing of resting-state data.

The principal difference between the processing of the modalities raised by reviewers was our approach for constructing the network matrices: we employed regularised partial correlations for the fMRI data, but ordinary (full) correlation for the MEG data. Different modalities do require different approaches, but the objective of taking these different courses is to identify bottom-line measures that are accurate and comparable across modalities. In short, the consistent approach we have taken is to use the best-practice inference methods in each modality.

We have adjusted our Material and methods section as follows, “We use slightly different network estimation methods in the two modalities. […] We did repeat our fMRI analysis using full correlation, rather than partial correlations, and found qualitatively similar results, with no difference in the significance of the claims we would make.”

Additional remarks on this revision:

Our revised submission uses the additional data available in the HCP900 data release. These new data (28 additional MEG subjects and 359 additional MRI subjects), together with the additional control variables included in our heritability analyses, and updated twin zygosities derived from genotyping data (rather than exclusively self-reported statuses) have led to slight changes in our results and parameter estimates, although not in any substantive way to our conclusions. We estimate lower values for heritability in all modalities (although within the previous confidence intervals), which may in part be related to the tighter control over noise confounds. However, the principal claim of our paper, that genetic effects are more important than the environmental factors that twins share, is now supported in the MEG beta-band, as well as in the fMRI analysis.

Overall, as a result of the input from reviewers, we now have even more confidence in the strength of the results we are presenting, in their generalisability and in their relevance. We thank the reviewers for their contribution, and set out below any remaining comments relevant to their more specific, individual points.

Reviewer #1:Although the authors show THAT genetics influences FC, they provide no information about WHY genetics influences FC. Which of the many mechanisms that contribute to observed BOLD FC are influenced by genetics?

Thank you for prompting us to include more comment about the plausible mechanisms mediating the genetic influence. We highlighted the extension of our discussion in part 1) of the responses above.

As the authors note, several of the results reported here (such as the heritability of FC from EEG data) have been reported already in previous studies. It is claimed here that these previous studies are less interpretable than the current study. That may be true, but then the authors need to provide the interpretation. If not then this paper loses a lot of its potential novelty and impact and I am not sure that it will be suitable for publication in this venue.

Thank you for highlighting the need to expand upon this point, as we feel that it is an important one. The relevant changes we made to our manuscript are described above in our response to the reviewers’ novelty concerns (1, above).

Some effort should be made to explain several of the more interesting and unusual effects as well. For example, it is stated that there is significant heritability in the alpha and beta bands of the MEG data but not in the theta band. Why? What is a plausible mechanism that would generate this pattern of results?

On this point, we would suggest that the lack of significant results in the theta band is not a reflection on the influence of genetics, but instead a product of the limited statistical power caused by the noisiness of estimating functional connectivity matrices in MEG. This is consistent with our past work (Colclough et al., 2016, NeuroImage), which demonstrated that all common methods for estimating resting-state connectivity in MEG show low repeatability, even over consecutive recording sessions from the same subject. The same paper showed a breakdown of this performance by frequency band, and revealed that network estimation was particularly noisy for all frequency bands other than alpha and beta. Additionally, the generators of theta power are more frontal in cortex than alpha and beta (Hipp et al.,2012, Nature Neuroscience), and the MEG scanner used by the HCP seems to have poor frontal sensitivity (and poorer sensor coverage in frontal regions), at least in comparison to the CTF MEG scanner. As a final note in this vein, our 2016 NeuroImage paper compared a wide variety of methods for inferring functional connectivity in MEG. The approach we use in this paper was chosen because it demonstrated the greatest repeatability in our method comparison.

We have slightly re-arranged the relevant part of our Discussion to make our thoughts on this point explicit, “We do not find significant genetic control of functional connectivity in the theta frequency band. […] Our claims are based on significant results from robust nonparametric tests, but the confidence intervals on our parameter estimates are inevitably broad.”

There is one major class of potential confounds which appear to be given short shrift here: potential contamination of the FC data by confounding factors having nothing to do with cortical connectivity or communication, but which are nevertheless heritable. The problem is that genetic factors might influence processes that are well known to influence correlations in MRI data (and likely MEG data as well). For example, if genetics causes some people to wiggle more in the magnet, that is obviously going to influence FC. (In fact I am fairly certain that I saw another study recently that reported just that effect, though unfortunately I am unable to find it now.)

We completely agree with this – that a major concern for genetic analyses of heritability is the wide array of possible co-varying confounds. Please see above for a more detailed response to these points (item 2).

It is possible that the paper being alluded to by the reviewer was Couvy-Duchesne et al., 2014, this seemed the most relevant authority, and we now cite it in our Discussion.

To take just one specific example of the above concern, it is reported that the visual system has a high degree of FC in the MEG data. The most likely candidate here would be eye movements, which will clearly affect MEG correlations, and which may also be influenced by genetics. However, I didn't see any report here that the eyes were tracked properly, or that the eye tracking data were regressed out when calculating FC. It is precisely these sorts of potential confounds (i.e., genetic influences operating on behavior that in turn influences FC rather than operating on FC directly) that must be scrupulously accounted for before publication.

Unfortunately, eye motion was not tracked as part of the HCP-protocol for the MEG resting-state scans, so we are unable to control for this directly. However, any independent components of the MEG sensor data that correspond to eye movements are removed during pre-processing. Further, while there is evidence that saccades and micro-saccades affect gamma-band activity, we are not actually measuring gamma-band functional connectivity, and there is not a strong evidence base for their impact on lower frequency recordings. Please see item (3) at the start of our response for how we have altered our discussion to reflect these points.

We did find strong connectivity in the visual system, particularly in the alpha-band – but this is well-reported in the MEG resting-state literature, and not thought to be just a product of eye movements (see, for example, Brookes et al., 2011, Baker et al,. 2014, Colclough et al.,2015, Hipp et al., 2012, Mantini et al., 2007, Marzetti et al., 2013, de Pasquale et al., 2012.

Perhaps one good way to get an intuitive handle on these sorts of potential indirect genetic influences would be to run separate analyses to determine the heritability of the various confounds that are known to affect FC, such as motion, eye movements, beamforming errors (in the MEG data). For that matter other factors that might affect FC indirectly might also be heritable. BOLD data taken directly off the scanner are non-Gaussian, and these are Gausianized during pre-processing by a necessarily imprecise procedure. Is the distribution of raw BOLD signals heritable? All these seem plausible, and all could potentially affect FC. A systematic analysis of these factors would seem to be critical for making any strong claims about direct genetic influences on FC.

The difficulty with performing separate analyses is that we expect many of these factors to be heritable in and of themselves. The real question is whether or not our functional connectivity estimates contain residual elements of these effects, or whether we are assessing the heritability of FC only, as we desire. For example, we have conducted additional heritability analyses on the power in each network node. In the fMRI data, we find significant heritability, on average, of these signal strengths (which is not necessarily surprising), and we must rely on our use of power as a confound regressor in the FC analysis to be confident in our claims. We believe we are the first authors to measure the heritability of FC that have tried to account for the effects of signal power in this way.

We have taken your suggestions about possible additional confounds for the MEG connectivity estimates, and included additional regressors for beamforming noise and for subject motion in our analysis. Please see the discussion on item 2) at the start of this response for details.

The claims in the Abstract and Discussion are a bit over-stated in several places given the results because they do not make any reference to places where no heritability is found. For example, while effects in the MEG alpha and beta frequency bands are observed, this is not true for the theta frequency band.

Thank you for highlighting this. We have explicitly discussed the lack of findings in theta (see above, in response to your third question), and altered our Discussion and Abstract to be specific about the bands where we find significant results.

The relevant part of the Abstract now reads, “On average over all connections, genes account for about 15% of the observed variance in fMRI connectivity (and about 10% in alpha-band and 20% in beta-band oscillatory power synchronisation),”

And that of the Discussion reads, “we estimate that the average heritability of individual connection strengths is about 15- 18% for BOLD correlations, and between 13% and 26% (1% and 25%) for correlations in beta-band (alpha-band) power fluctuations.”

The figures and the supplementary movies in this paper are rather poor, especially the connectivity figures. There are many excellent choices for visualization software these days.

We have replaced the connectivity figures with improved versions, but have distinguished between the principal figure of the paper (Figure 1), and the supplementary figures.

Particularly, we have altered the connectivity plots in Figure 1 to an alternative that has more visual impact and clear separation of the network nodes. We have retained the glass-brain style figures in the supplementary material, as we find it aids our understanding of the networks to see the connections in the context of the cortical locations of the nodes. This compromise offers a balance between visual aesthetics and ease of comprehension.

The entire data analysis procedure is based on the HCP ICA parcellation. It would inspire more confidence in the conclusions if some analysis was provided that showed whether the choice of parcellation scheme makes any difference on the results or the conclusions. I don't think that it would make much difference in the results if one compared the HCP ICA parcellation to, say, the Glasser HCP parcellation, because these are both fine-scale parcellations that are likely to include more spatial information than can really be supported by genetics data anyway. (In the case of the MEG data, the HCP ICA parcellation may very well have higher resolution than can be supported by either the genetics or the MEG.) It would be immensely helpful to know whether a coarser parcellation schemewould change the results, and I think that knowing the answer to this question might also help us address the question of why these apparent genetic influences are found.

Thank you for this discussion. We agree that an understanding of how parcellations may affect the heritability of connectivity is important, and past work has not touched upon this point. We also agree that a denser parcellation scheme would not be a sensible comparator, for the reasons you mention. We repeated our analysis with the 15-dimensional HCP ICA parcellation, which has a substantially different structure to the 39-dimensional parcellation we originally used. As your point was raised by several reviewers, we discussed our approach and findings in item 3) at the start of this response.

It looks as if subject motion was removed during pre-processing in both the fMRI and the MEG data, but subject motion was still used as a regressor during ACE modeling in the fMRI data. Subject motion is a problem in both fMRI and MEG, so if it was a problem in the fMRI data even after pre-processing why was pre-processing sufficient for the MEG data? At a minimum it seems that it should also be included in the MEG modeling analysis.

We take your point that motion is an important confound to treat carefully.

It is worth noting that the pre-processing for motion is slightly different between the modalities. The fMRI pipeline registers functional volumes into the same static space, then uses an automated classification system to remove ICA components from the data associated with motion (and other structured noise) artefacts. Additionally, a variable summarising the motion of the subject during the recording can be used as a regressor in further analyses. On the other hand, the MEG pre-processing registers the signals in the sensors into a common, static reference frame before source reconstruction, followed by an ICA decomposition process for further noise removal. No summary measure of motion in the scanner is available (though see below). These approaches represent the current best practice in pre-processing functional images for motion in these modalities.

In the absence of a MEG-specific motion confound for the resting-state scans, we have now used the summary measure from each subject’s fMRI scans as a proxy motion variable for the MEG analysis. To the extent that motion is heritable, this proxy measure is the best available approach for removing any residual motion confounds in this dataset. The changes to our text were highlighted at the start of our response, in part 2).

This is a side point, but I find it surprising that it is useful/helpful to regress out the MR image reconstruction software version in ACE modeling. This suggests that these data really do live at the very limit of sensitivity, and that confounds and bias really can creep into the data even after pre-processing. It seems like the best practice would be to use the same version of the software to process all the data and I find it a bit worrisome that this was not done.

The reconstruction of raw (complex, multi-coil, multiband, k-space) fMRI data, was carried out by the HCP. This is outside our control – and indeed at this point theirs – as it was not possible for practical reasons to store these raw data long-term. However, we have no reason to think that, after inclusion of this confound variable, any signature of this change remained in the data. A full explanation of this change is given on the HCP website: https://wiki.humanconnectome.org/display/PublicData/Ramifications+of+Image+Reconstructio n+Version+Differences

In Subsection “E. Three-component variance models” it is stated that the variance partitioning analysis addresses non-negative portions of the variance. Why do these procedures produce negative variance estimates at all? Is this just statistical error?

We apologise for the potentially confusing wording; we had written, "This model ascribes non- negative proportions of the variance either to additive shared genetics […]” Of course, variance is always non-negative, and the model appropriately constrains the variance parameters. This wording was to reference that we allow h^2^ and c^2^ to be exactly zero. On reflection, this is self- evident, and so we have simply dropped the word “non-negative.”

Reviewer #2:Colclough and colleagues examine the genetic influences on resting state MRI and MEG connectivity in "The heritability of multi-modal connectivity in human brain activity." Subjects included 461 individuals with MRI data and a group of 61 primarily overlapping individuals with MEG data all from the Human Connectome Project. Subjects were from extended twin pairs or unrelated individuals. The goals of the article are to estimate the heritability and common genetic influences on resting state and MEG measures of connectivity. The article is clearly written and the analytic plan is both complex and reasonable. Unfortunately, the findings represent only an incremental increase in scientific knowledge, as the genetic influences on resting state have been reported several times using conceptually similar methods. While the MEG results are novel, that analysis is rather underpowered, as the authors appropriately note in the Discussion. Thus, while I believe the findings should be reported, the authors should either describe the work as replication or conduct additional analyses that will increase novelty.

Thank you for this overview. We have discussed all reviewers’ concerns over novelty at the start of this response, in item 1). This includes changes to better contextualise our findings and the way in which they advance the state of knowledge.

Reviewer #3:1) I think the authors should clarify why 39 cortical regions are chosen, as opposed to other common parcellation schemes, which often have 50-100 regions.

Larger parcellation schemes (>60 brain regions) are unfortunately impractical for MEG: the scanner only records from several hundred sensors, and the effective dimensionality of the data, once it is localised within the cortex using established methods, is on the order of 60. Models that separate sources that are internal and external to the MEG dewar tend to provide support for 60 sources or fewer (Taulu et al., 2005), and recently developed adaptive data driven parcellations that combine MEG *and* EEG data only find that there are on the order of 70 unique parcels identifiable [http://biorxiv.org/content/early/2017/01/04/097774]. In our experience, ~40 regions is a good compromise for a parcellation that can represent contiguous nodes, but does not stretch beyond the support of the spatial information available in the MEG data. Note that we have, however, included an additional analysis on a lower, 15-dimensional cortical parcellation; please see item (3) at the start of our response for a more extended discussion of parcellation schemes and the results of this analysis.

We have included this paragraph in the Material and methods, to provide additional explanation, “This parcellation was chosen because the use of contiguous, focal nodes creates a comprehensible network model between individual regions. […] This upper bound is supported by models that categorise signals as originating within or without the MEG dewar, which suggest a limit of about 60 measurable cortical sources (Taulu et al., 2005), and by recently developed data-driven parcellations that only find around 70 unique identifiable parcels using combined MEG and EEG data (Farahibozorg et al., 2017).”

2) I'm surprised that the Authors did not at least attempt to reconstruct networks in the higher gamma band. Why only theta/alpha/beta?

For two reasons. First, in our previous work (Colclough et al., 2016) we assessed the reproducibility of network estimation in resting-state MEG. We found that by and large, reproducibility was low, and that network estimation is therefore noisy. The results were bad for all bands other than alpha and beta, where most structured signal originates in the resting state. Secondly, and this is the reason that the theta band was included at all, because the evidence in the literature is that resting-state networks present in MEG primarily in the theta, alpha and beta bands, but less clearly in lower or higher frequencies [Hipp et al., 2012]. This does not necessarily mean that there is not structured connectivity between oscillations in these other bands (indeed, ECoG studies find strong relationships between gamma oscillations and fMRI resting-state networks; see for example Nir et al., 2008), but simply that these relationships do not emerge in MEG data using conventional functional connectivity techniques for MEG. We now refer to this in a sentence in the Introduction, “These bands span the frequency range within which the most convincing patterns of resting-state MEG connectivity have been shown to be expressed (Baker et al., 2014; Brookes et al., 2011a; Hipp et al., 2012; Mantini et al., 2007; Marzetti et al., 2013; de Pasquale et al., 2012, 2015).”

3) One of my main concerns is the use of partial correlations in the FMRI analysis and simple correlations in the MEG network analysis. Why this difference? The Authors cite two of their own papers as justification but this not particularly convincing. Could we have a short justification here?

We agree that the original presentation of our methodology should have included more explicit discussion of some of these choices. We have addressed the reviewer’s concerns on this point at the start of our response, under item 4).

4) The authors state that the beta-band "exhibits broad connectivity over the whole cortex". That doesn't really appear to be the case looking at Figure 1. In ay case, what does that statement actually mean – it's quite a woolly phrase that is not backed up by any kind of quantification.

Thank you for highlighting this. We have removed the poor phrasing, and now describe the group-mean beta-band connectivity matrices as, “The beta-band exhibits strong bilateral coupling across the sensorimotor cortices, with connectivity continuing through the superior parietal lobes and down to occipital cortex.”

5) The authors look at both genetic and environmental factors in terms of contribution to signal power (in both FMRI and MEG), partly as a control for these as confounds in the connectivity analyses, but also as interesting exploratory analyses in their own right. I have two concerns here. 1) Can the Authors please describe how "power" was assessed in the FMRI signal? I assume this is some measure of temporal variation around the mean of the voxel time series? 2) For the MEG, the power passed by a beamformer in an RSN analysis can be quite dependent on geometric effects (even after weights normalisation). Wouldn't a better measure of 'activity" in the MEG amplitude envelopes be some measure of temporal variability? The standard deviation (SD) is often used, but the SD of the virtual-sensor and the Mean are often correlated in beamformer reconstructions, so a better measure might be some proportional change (e.g. SD/Mean). "Activity" in both the FMRI and MEG time series could thus be assessed using the same (or very similar) metric.

We agree that we had neglected to mention how this was calculated. We had in fact used the standard deviation of the node time courses (fMRI signal or MEG power envelope) as our measure of “power,” as suggested. We have now explained our approach in the text. We have also followed the reviewer’s advice, and included an additional power confound in our MEG genetic analyses that is the coefficient of variation (SD/mean) for each node. We have highlighted the altered text in item 2) near the top of this response, where we address our use of additional confounds.

6) In the ACE model, quite a few nuisance covariates are regressed out of the model before heritability/environmental effects were assessed. How were these parameters (and their second-order versions) chosen? In addition, when regressing out some many parameters, is there an issue with statistical power in the MEG analyses as some of the groups only have relatively few subjects (i.e. 11 monozygotic twin pairs)?

As gross brain size is known to vary greatly by gender and age, it is routine in brain imaging studies to remove effects of age, gender and, when the sample size supports it, age^2^ and interactions between these variables. Most genetic analyses will also include regressors of this form to control for differences in basic demographics between unrelated subject pairings. The MEG sample we originally used totaled 44 twins; while of course a larger sample is desirable (and we have now increased it to 64), inaccurate fitting of age^2^ and age*gender interactions will only degrade our residuals and reduce our ability to detect heritability. Hence, we see this as a conservative approach (and it retains symmetry with the use of these regressors in the larger fMRI sample).

7) Finally, with these "nuisance" regressors Is there a potential problem with interaction with heritability? For example age is 100% matched in the twin-pairs, but is presumably not matched for the non-related pairings.

It is precisely because the lack of age matching between siblings that these nuisance regressors are essential. We have considered this carefully, and have not been able to construct a setting where use of covariates (which remove variance) can artificially inflate heritability.